# Elevator Technology Improvements: A Snapshot

## Kheir Al-Kodmany 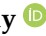

Department of Urban Planning and Policy, University of Illinois at Chicago, Chicago, IL 60607, USA;
kheir@uic.edu

**Abstract:** Efficient vertical transportation is vital to a skyscraper's functional operation and the convenience and satisfaction of its tenants. This review complements the author's previously published research by updating the readers on innovative hardware and software-based solutions. It lays out, organizes, and combines extensive and scattered material on numerous aspects of elevator design in a straightforward and non-technical narrative. Rope-less elevators, the MULTI, artificial intelligence (AI), the Internet of Things (IoT), and extended reality technologies are some of the developments and advancements this article examines. The analysis also contextualizes current technical developments by reviewing how they are used in significant projects such as the One World Trade Center in New York City. Lastly, the paper examines innovative technologies, such as holographic elevator buttons and ultraviolet rays that disinfect elevators, in response to the COVID-19 pandemic.

**Keywords:** basic systems; advanced systems; long distances; smart solutions





## 1. Introduction

### 1.1. Background

One hundred fifty years ago, cities appeared significantly different than they do today. Often, the cityscape was flat and uniformly patterned. The heights of residential and commercial structures were rarely as tall as flagpoles. However, today, cities are growing vertically. Population increases, rapid urban regeneration, rising land prices, active agglomeration, ego, and globalization drive building upward. Indeed, the race to build the world's highest skyscraper seems to go on forever, reaching ever-impressive heights. Around the beginning of the new Millennium, in 1998, Kuala Lumpur, Malaysia, built the 452 m (1483 ft) Petronas Towers, snatching the title of the world's tallest building from the 442 m (1450 ft) Sears Tower (renamed Willis Tower) constructed in 1973 in Chicago. In 2004, Taipei, Taiwan, erected the 508 m (1667 ft) Taipei 101. In 2010, Dubai, UAE, built the 828 m (2717 ft) Burj Khalifa, the world's tallest. As such, in just 12 years, the height of the tallest building almost doubled [1,2].

Besides globalization and land prices, the rapid increase in urban population forces cities to build upward. The United Nations predicts that 70% of the world population in 2050, about 9.7 billion, will live in urban areas, up from 51% in 2010. Such an increase entails adding almost a quarter million urban dwellers globally every week. To expand cities horizontally to accommodate urban population increase, we will face sprawl problems. Sprawl has caused numerous economic, social, and environmental crises and is an unsustainable way to grow. After learning the hard lessons of urban sprawl ills, planners have reverted to the vertical and compact model [3]. As such, since the turn of the century, many cities have been erecting high-rises worldwide.

High-rise buildings consume more energy than low-rise buildings for many reasons, including the employment of vertical transportation. Elevators use between 5 and 15% of a high-rise building's power, so efforts to reduce their energy consumption are worthwhile. Further, elevators use significant valuable space in a skyscraper. Sometimes, they may occupy 25–40% of the floor plans. Of course, this figure includes all elevators (e.g., passenger,

freight, emergency, and shuttle). Therefore, reducing the required space and number of shafts is a sought-after goal in elevator design [1].

*1.2. New Elevator Systems*

Like automobiles and rail transport, elevators are becoming increasingly high-tech, pushing manufacturers to improve elevator speed and safety. The development of elevator speed has been astounding. If we compare the speed of the first passenger elevators (12 m per minute) to that of the world's fastest elevators (1200 m per minute) located at the CTF Finance Center in Guangzhou, China, the speed has increased one hundredfold. For improving passenger flow, destination dispatching systems are most efficient. When passengers click buttons corresponding to their desired floors, the system directs them to the elevators with the shortest travel times. Enhanced routing will result in more efficient passenger transfer, especially during peak traffic in hotels, residences, and offices. Further, new systems allow building managers to program elevators to correspond most efficiently to passengers' demands throughout the week, day, night, and holidays.

As time passes, elevators are becoming more intelligent and safer. Modern elevators provide smooth, comfortable journeys for passengers while covering greater distances, reducing the need to use transfer or sky lobbies. They are also energy-efficient; some produce energy, such as the regenerative drive. New design promises to make elevators move not only up and down but also sideways and diagonally. Such innovative design will revolutionize the architecture and layout of high-rise developments. It will allow buildings to achieve more excellent connectivity and improve people flow.

However, building elevators are not an exception to the harsh reality that everything eventually wears out and must be replaced. Even with routine maintenance, old machinery always requires updates. Modernizing elevators is a feasible way to increase the value and appeal of a tall building. Intelligent elevator systems provide enhanced travel comfort and the flexibility to adapt to changing building requirements, thereby enhancing performance [4,5].

Internet-connected elevators represent the cutting edge of elevator maintenance. This technology notifies building managers in real-time when a problem is beginning to develop. This is intended to reduce maintenance expenses and save time. Sensors gather data on variables, such as usage, that can impact the deterioration of components. The data are sent to a cloud-based platform for processing and analysis, enabling building managers to apply proactive measures, preventing problems from occurring.

*1.3. Goals and Objectives*

Any elevator design and production improvement will significantly impact expenditures, customer satisfaction, and the natural environment. This article explores contemporary advancements in the elevator sector to educate the wider community of engineering and architectural students and professionals. It explores the development of elevator technology and considers how it affects their construction and upkeep. It compiles and organizes detailed and fragmented knowledge on various elevator design topics in a clear and non-technical discourse.

The specific goals of this paper are to answer important questions concerning elevator design and development as follows:

- What are the essential components and operational systems of an elevator?
- What are the new advances in elevator systems?
- What are project examples of mega- and supertall buildings that incorporate sophisticated elevator systems?
- What are the retrofitting and modernization options for elevators?

Overall, this paper aims to simplify complex engineering concepts and make them accessible to a broad audience, a fundamental objective of the *Engineering Encyclopedia*.

## 2. The Basics

### 2.1. Notes on Elevator History

Humans have used hoists for vertical transportation since the third century BCE. The ancient Greeks allegedly invented elevators using pulleys and winches. However, modern elevators, such as those we know today, were developed in the 1800s. In the 1850s, elevator design shifted from carrying goods to carrying people, and the first elevator safety devices were invented in 1852. The development of safety devices paved the way for the advancement of passenger elevators [6].

In 1857, the first passenger elevator was constructed. This elevator was steam-powered and moved at a leisurely 12 m (40 ft) per minute. Early passenger elevators were considered more of a spectacle and luxury experience than a mode of transportation. Because tall buildings were not yet built, higher floors had lower rents due to the need to climb stairs to access these units—a stark contrast to today's high-priced penthouse units.

In the 1870s, elevators evolved from a novelty experience reserved for a select few to a transportation staple. As buildings became taller and taller, increased elevator speeds became the driving force behind elevator innovation. The first elevator-equipped office building was built in Downtown Manhattan in 1870, marking the beginning of widespread, practical elevator history.

As mentioned, elevators were initially powered by steam. However, as they were installed in office buildings and speed became a priority, they gradually evolved to hydraulic power. Around this time, the industry standard of 30 s or less wait times was established, and it is still the industry standard today [6].

Elevator safety and speed advanced even further when electricity became the primary power source for elevators. Elevator shafts had become an integral part of architectural design by the 1880s as buildings rose in height. Hence, in 1880, the first electric elevator was built, and in 1887, automatic doors were produced that locked off the vehicle from the shaft, making elevators a safer mode of transportation for passengers. In a commercial building in 1889, the first successful electric elevator was installed.

In 1902, the Otis Elevator Company created the electric traction elevator, a significant breakthrough. This innovation ingeniously integrated the electrical and mechanical subsystems of an elevator. In 1904, the first elevators of this type were installed in a commercial building in New York City. Vertical transportation became safer and more convenient than ever before as elevators evolved and expanded to include more advanced safety and efficiency features [6].

### 2.2. Essential Terminologies

Essential terminologies must be clarified before diving into elevator mechanical and electrical systems [1,2,4–6].

**Hoistway**. This is the vertical enclosure that runs through a structure and is sometimes referred to as an elevator shaft. Within this enclosure an elevator car travels.

**Car**. This is the vehicle for transporting people and goods vertically within a structure. The car moves along the hoistway, where it is directed by rails at the top and bottom and propelled vertically by a mechanism that operates an elevator. It is also commonly referred to as a cab.

**Pit**. This is the area inside a hoistway between the elevator's lowest floor landing and the hoistway floor. The pit offers the space required for the elevator to work effectively and safely while also acting as a storage location for important elevator equipment. It has four walls of concrete or concrete blocks and a concrete base slab.

**Travel Height**. This is the vertical distance an elevator travels between the building's lowest and highest floors. This is analogous to the number of floors serviced, except that the height from floor to floor varies from one building to another. Additionally, floor heights could vary in the same building. As a result, the actual height traveled is a more accurate marker than the floor counts.

**Overhead**. A space within a hoistway lies between the highest-level landing that is serviced by the elevator and the ceiling of the hoistway. It is vital to have sufficient headroom to accommodate the elevator's machinery and supply the space required for the elevator to operate securely and reliably.

**Bank**. A cluster of two or more elevators controlled and dispatched by a single or multiple call-button panel(s).

**Elevator System**. The elevator system should comply with the building code requirements. It should also integrate with other systems, including architectural, structural, electrical, mechanical, plumbing, technological, and fire protection.

**Lobby Floors or Sky Lobbies**. These are intermediary floors of a skyscraper where passengers switch from an express elevator to a local elevator and vice versa.

**Wait Time**. This is the period between calling the elevator and its arrival.

### 2.3. Elevator Types

In tall buildings, different basic types of elevators are often employed [4–6].

**Passenger Elevator**. A passenger elevator's primary purpose is to transport people or lightweight goods. Conventionally, passenger elevators are calculated for weight loads of between 907 kg (2000 lb) and 2268 kg (5000 lb); however, some models are available with higher load ratings. These weight measures correspond to car areas that range between 2.3 and 5.1 m$^2$ (25 and 55 ft$^2$). Therefore, these elevators usually take lighter loads and are smaller than freight elevators.

**Freight Elevator**. These are designed to transport heavy goods and equipment. They are characterized by doors that open vertically (such as cage doors), and they are robust but have unsightly finishes. Individual buttons are necessary for freight elevators to call the car and open and close the doors. Yet, a freight elevator is less automated than a standard elevator, so the user has less control over how the elevator works.

**Service Elevator**. These elevators carry both people and light equipment simultaneously. Not to be confused with freight elevators, the shape of service elevator cars is deep and narrow, making it easier to load long cleaning carts, supply carts, and similar lightweight equipment. In contrast, passenger-shaped elevator cars are often shallow and wide, making it easier for passengers to enter and exit the front of the elevator without becoming trapped in the back. In addition to the front door, rear doors are available on both passenger and service car designs.

**Shuttle Elevator**. The shuttle elevator travels rapidly between lobby floors. To reach their desired floor, travelers switch to a local elevator on the lobby level. Connecting high-capacity/high-speed shuttle elevators with local elevators makes traveling efficiently throughout a tall structure possible. Not only do the elevator/shuttle systems save space, but they also save time.

**Emergency Elevator**. These elevators are installed so that fire crews can reach the upper floors to conduct necessary rescue procedures and evacuate trapped tenants. Since other elevators (including passenger elevators) are shut during a fire incident, emergency elevators are paramount. They are also crucial for disabled people. During fire incidents, tenants with mobility impairments are evacuated via emergency elevators with the assistance of firefighters.

### 2.4. Leading Elevator Companies

The booming market for elevators is fueling innovation. Table 1 displays the world's leading elevator companies. These businesses are all involved in research and international trade and often report significant annual revenues. Due to its 160-year-plus service, profundity, and successful multi-brand strategy in the global market, Otis maintains a commanding lead over other manufacturers. Otis dates back to 1853 when Elisha Graves Otis launched the first passenger elevator with safety features at the Crystal Palace Conference in NYC. After three years, the first passenger elevator was installed in the same city. With its consistent and robust growth in recent years and multi-brand approach in Asia,

Schindler continues to hold a prominent position among elevator manufacturers. KONE and ThyssenKrupp are also pioneering elevator manufacturers, constantly offering the market innovative solutions. Japanese brands are also providing cutting-edge technologies and are most popular in Asia.

**Table 1.** Leading Elevator Companies. (Source: compiled by author).

| | Company | Country |
|---|---|---|
| 1 | Otis | USA |
| 2 | Schindler | Switzerland |
| 3 | KONE | Finland |
| 4 | ThyssenKrupp | Germany |
| 5 | Hitachi | Japan |
| 6 | Mitsubishi Electric | Japan |
| 7 | Toshiba Elevator | Japan |
| 8 | Fujitec | Japan |
| 9 | Hyundai Elevator | Korea |
| 10 | Canny | China |

*2.5. Main Elevator Systems*

There are two main types of elevator systems: hydraulic and traction. Hydraulic elevators obtain power from hydraulic jacks, fluid-driven pistons inside a cylinder. In contrast, steel ropes or belts are wound around pulleys and used to operate traction elevators (Figure 1).

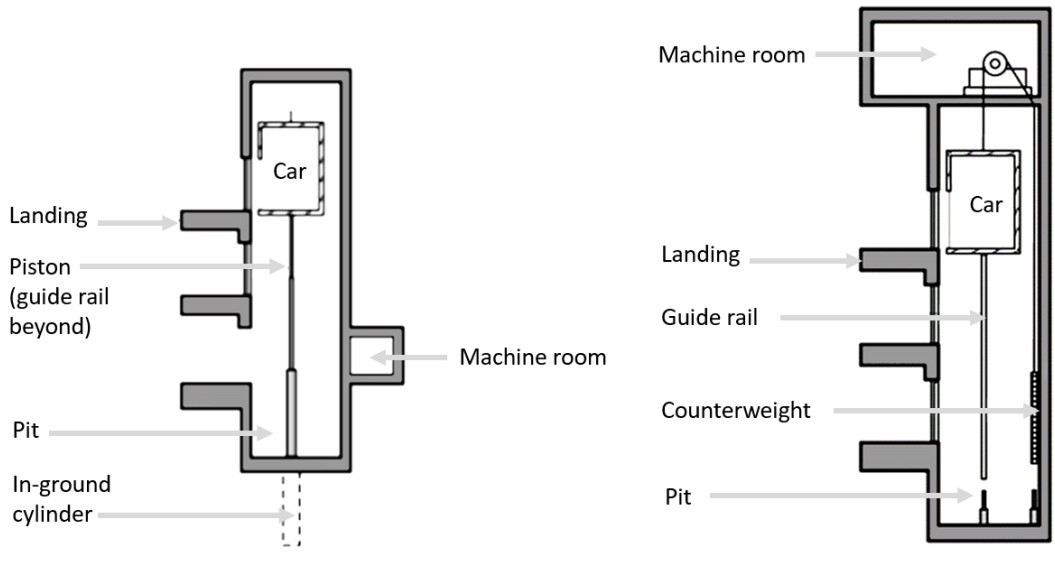

**Figure 1.** A schematic illustration of the differences between hydraulic and traction elevators. (Source: author).

2.5.1. Hydraulic Elevators

Throughout the second half of the twentieth century, hydraulic elevators were the most common type of elevator mechanism used in buildings. The hydraulic elevator car is propelled higher by a piston, driven upward by a pumping mechanism that pushes hydraulic fluid into the piston. In most instances, hydraulic elevators will need a small

chamber of around 7 m$^2$ (75 ft$^2$) located close to the hoistway to house either the pumping or control units [5]. Hydraulic elevators, unlike traction elevators, do not require overhead hoisting machinery. Their advantages include inexpensive installation, reasonable maintenance and repair costs, and reliability for hauling heavy loads. However, hydraulic elevators can reach a maximum speed of 46 m (150 ft) per minute. Because of their lower velocity, they are often not recommended for tall buildings.

The term "holed" or "hole-less" describes the position of the piston in hydraulic elevators, which determines which type of elevator it is. As such, we differentiate among three types of hydraulic elevators:

1. A traditional hydraulic elevator with a sheave that extends below the elevator pit's floor. The pit holds the retracting piston as the elevator lowers. It has a maximum travel range of 18 m (60 ft).
2. Hole-less hydraulic elevators are comparable to regular ones, except they do not need a sheave or hole in the ground. Compared to a non-telescopic piston, the telescopic piston design provides a travel distance of up to 15 m (50 ft).
3. In the case of the roped hydraulic elevator, the car is propelled by both ropes and pistons. It can move a maximum of 18 m (60 ft) in one direction.

### 2.5.2. Traction Elevators

Lifting in traction elevators is accomplished by ropes that travel across a wheel driven by an electric motor. The wheel is set in motion by the beginning of the electric motor's rotation. This pulls the rope, which raises the elevator vehicle to the appropriate building levels. The machine room on the highest floor of the building is typically where the wheel for this setup is kept because this is where it is most secure. The elevators are given a counterweight to make them more efficient by countering the weight of the car and the people riding in it. As such, this system facilitates coordination between the speed of the wheel and the rope [6].

There are two distinct varieties of traction elevators:

1. Geared Traction Elevator: This type of traction elevator has a gearbox coupled to the motor to drive the wheel. These elevators can travel at a speed of up to 152 m (500 ft) per minute. The geared traction lifts have a maximum travel distance of 76 m (250 ft).
2. Gearless Traction Elevator: With this form of elevator, the wheel is connected to the motor directly. It has a maximum travel distance of approximately 610 m (2000 ft) and can move at a maximum speed of up to 610 m (2000 ft) per minute. It is the most popular option for use in tall structures [1,2,6].

### 2.5.3. Machine Room Elevators

In machine room elevators, the traction machine and control equipment are stored in a separate chamber, which is often situated above the hoistway. The machine room offers good maintenance access, preserves equipment temperature ranges, and enables flexibility to suit most performance needs. Because other types of elevators have a restricted reach, machine room elevators are often used in tall buildings with more than 25 stops and can travel longer distances. The two primary disadvantages of machine room elevators are the costs of providing the machine room and the potential negative impact on the building's aesthetics, mainly impacting penthouses [6,7]. Machine room traction elevators are recommended for traveling 25 or more floors, while machine room-less traction elevators are recommended for traveling up to 25 stories (see next section).

### 2.5.4. Machine Room-Less (MRL) Elevators

One of the most critical advancements in elevator design since the first electric elevators were released a century earlier was MRL technology. Debuted in the mid-1990s, manufacturers eliminated the need for a machine room by downsizing the motors and other apparatus to fit inside the hoistway. Consequently, MRL is an elevator system that requires less space [6–8] (Figure 2). Additionally, MRL uses 30–40% less energy than comparable-

sized conventional traction and hydraulic motors. MRL's reduced power consumption also results from its lower starting current need. MRL and the regenerative drive together significantly improve energy efficiency (see next section).

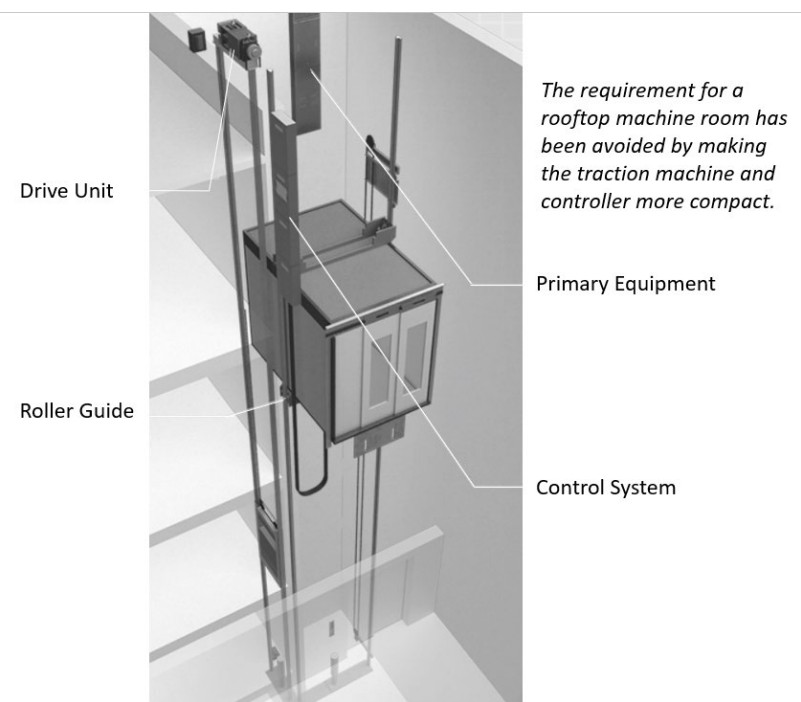

**Figure 2.** Machine-Room-Less (MRL) Elevator System. (Source: author).

### 3. Recent Developments and Advances in Elevator Systems

#### 3.1. Regenerative Drive

The regenerative motor, which enables "heat" energy to be recycled rather than squandered, is a critical advancement in energy-efficient elevator technology. It works by gathering and transforming the energy lost during braking, which is necessary to maintain the elevator's speed. In other words, it converts the mechanical "heat" energy generated by braking into electrical energy ("regenerated"), which is then transferred to the building power system. An elevator applies a brake in three cases: (1) when it goes up with a load of passengers lighter than the counterweight, (2) when it goes down with a load of passengers heavier than the counterweight, and (3) when it goes down with a load of passengers lighter than the counterweight. In these cases, the motor functions as an energy generator, converting mechanical energy into electrical energy and channeling it back into the facility's electrical system. Another elevator or other electrical devices can use the generated power. Importantly, these little quantities of energy generated during the brakes build up to significant savings over time. Regenerative drives can lower the energy consumption of a building transportation system by up to 70% [8,9]. The energy needed to run the HVAC system is also reduced because the structure and machinery exposed to the excessive heat produced by conventional motors no longer need to be cooled in the new system [10]. Despite the higher costs of the generative drive model, more buildings are embracing it due to its energy efficiency.

#### 3.2. Elevator Rope

Because it links the elevator motor with the cab, sheaves, and counterweight, the elevator rope is crucial to traction elevators. The car-lifting ropes are tied to a counterweight on the other side of the sheave. At around 40% of its capacity, the cab is balanced with the counterbalance. As a result, energy is conserved. Steel ropes hold cabins. However, these ropes can no longer support their own weight in very tall buildings since they become too

long and thick. Energy consumption increases with height as the initial currents and energy requirements rise. In response, elevator manufacturers have started making stronger cables. Aramid fiber rope, developed by The Schindler Company, is more durable and lightweight than a traditional steel rope. Otis created the Gen2 elevator that features polyurethane-coated ultra-thin wires replacing steel ropes. Mitsubishi used concentric steel wire to create a stronger, denser string. These stronger, lighter wires efficiently move elevator cabs using less energy [11,12].

KONE's UltraRope is among the most notable innovations. KONE is an elevator engineering company headquartered in Espoo near Helsinki, Finland. UltraRope enables elevator cars to travel 1000 m (3280 ft) in one run due to its carbon-fiber core and special high-friction coating. This is twice the typical maximum of 500 m (1640 ft). Notably, the 90% reduction in rope mass results in significant energy savings. UltraRope weighs 2500 kg for a 2000 kg (4409 lb) elevator traveling 500 m, as opposed to 27,000 kg (59,524 lb) for ultra-high tension steel ropes. KONE UltraRope elevators will be installed in the 1000 m (3281 ft) Jeddah Tower in Saudi Arabia (construction is on hold). In one of the tallest residential buildings in Europe and the first to be outfitted with KONE UltraRope, South Quay Plaza in London, UK, received eight of these elevators. The Marina Bay Sands in Singapore also utilizes the UltraRope [13,14].

### 3.3. Double-Deck Elevators

The requirement for fewer elevator shafts increases with skyscraper height since they take up valuable interior space on each story. Two stacked cabs make up a double-deck elevator, one serving levels with even numbers and the other odd. As such, the best use for double-deck elevators is in tall buildings, especially for shuttle services [14,15]. The double-deck elevators, however, have specific operational issues. Visitors to the structure are initially perplexed because each cab only has access to even or odd-numbered floors. Often, visitors need to take an escalator to the second lobby level to go up from the ground floor, for example. Second, both cabins must come to a complete stop, even if only one floor needs to be reached. Third, distances between floors must be the same. To solve the third problem, Toshiba created height-adjustable double-deck elevators, which give architects more flexibility when employing double-deck elevators. ThyssenKrupp has created newer technology to help in solving other problems. ThyssenKrupp is an engineering and steel production company headquartered in Essen, Germany. It developed the TWIN system, which enables two elevator cabs to function independently in the same shaft. The company believes that this innovation can significantly decrease the number of elevator shafts needed and, as a result, improve the amount of leasable space (see the following section) [15,16].

### 3.4. The TWIN System

The TWIN system was invented in 2003 to provide passengers and building owners greater efficiency, flexibility, and convenience. It enables two cabins to move independently in one shaft, saving 30% space and reducing the footprint by the same amount. Therefore, the TWIN is an elevator system with two standard cabs installed within the same shaft but differs from double-deck elevators in that these two cabs operate independently. The two cabs are kept at a safe distance from one another by a gadget that measures the distance. An automated system assigns passengers to cabs most effectively, reducing wait times and empty journeys while conserving energy [16]. This optimizes travel for both cabins. As previously indicated, the system enables the integration of more elevators within fewer shafts; it is predicted to need one-third fewer shafts than traditional elevators, freeing up valuable core space. Additionally, the TWIN system minimizes the building materials required for shafts, reducing embodied energy. Significant power and space savings are made possible because both elevators are controlled by a single machine in the same shaft [17,18]. ThyssenKrupp has been a trailblazing company in the creation of this ground-breaking system.

## 4. Case Study: One World Trade Center, New York City, NY, USA

Advanced elevator systems have been installed in significant skyscrapers. Examples include Burj Khalifa in Dubai, UAE; One World Trade Center in New York City, USA; and Shanghai Tower in Shanghai, China. The best-case scenario would be to review each of these project examples. Yet, because there is no space to discuss all of them in one article, the author must choose one and elaborate on it. Due to its symbolic significance to the United States, One World Trade Center received more attention throughout its design and construction than any other skyscraper. Thousands of workers, citizens, experts, engineers, and architects assisted in bringing this structure to the cutting edge of technology. In addition, the building's owners and authorities wished to use it as a model for future construction [1,2,19].

### 4.1. Building Overview

Standing at 541 m (1776 ft) tall, One World Trade Center is the highest building in the Americas (Figure 3). It was built in 2014 and was designed by SOM (Skidmore, Owings and Merrill). It has 104 stories above ground and 2 below. The tower, which serves as the property's focal point, is located on the 16-acre site where the Twin Towers once stood before the tragic events of September 11, 2001. It integrates cutting-edge environmental and green features, such as advanced elevator systems, and establishes new architectural and safety standards. The upgraded life-safety systems go above and beyond what the New York City Building Code requires. The tower is resistant to explosions, storms, and earthquakes due to using over 50,000 tons (35,714 cubic yards) of steel and 252,000 tons (180,000 cubic yards) of concrete for constructing the tower. Special safety precautions include pressurized and extra-wide emergency stairs, fireproofing, and air-filtration systems for chemical and biological particles. According to the Emporis database, the building is among the most expensive in the world; One World Trade Center had a total cost of USD 3.9 billion [20,21].

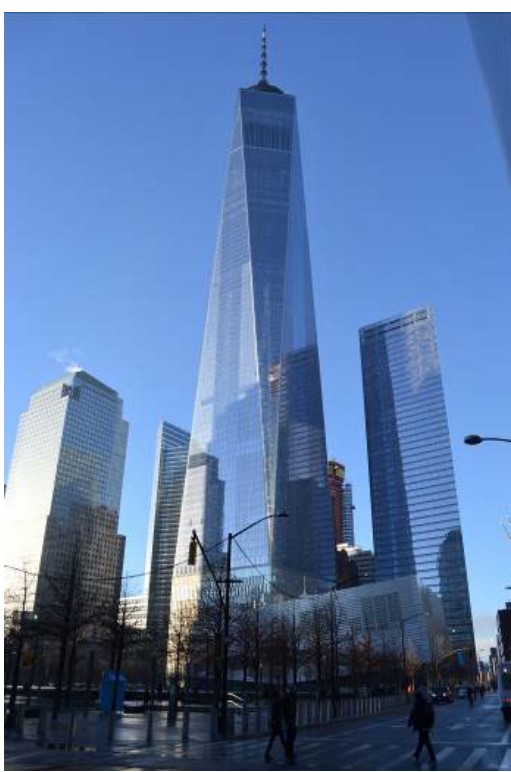

**Figure 3.** Standing at 541 m (1776 ft), One World Trade Center in NYC, USA, is the tallest building in the Western hemisphere. (Photo by author).

*4.2. Elevator Systems*

Five of the One WTC's 73 elevators are express, with a top speed of more than 36.5 kph [10.16 m/s (33.3 ft/s)]. These express elevators are the quickest in the Western Hemisphere and can carry 1814 kg (4000 lb). As a result, it takes around 40 s to go the 394 m (1293 ft) leading to the observation deck, which is on the 102nd floor (Figure 4). Elevator speed was increased from 9.1 to 10.16 m/s to accommodate the expected large number of guests visiting the observation deck at One World Trade Center. The design team anticipated 10,000 people working daily on the office floors and over 5 million visitors per year (14,000 per day) to the observation deck [22].

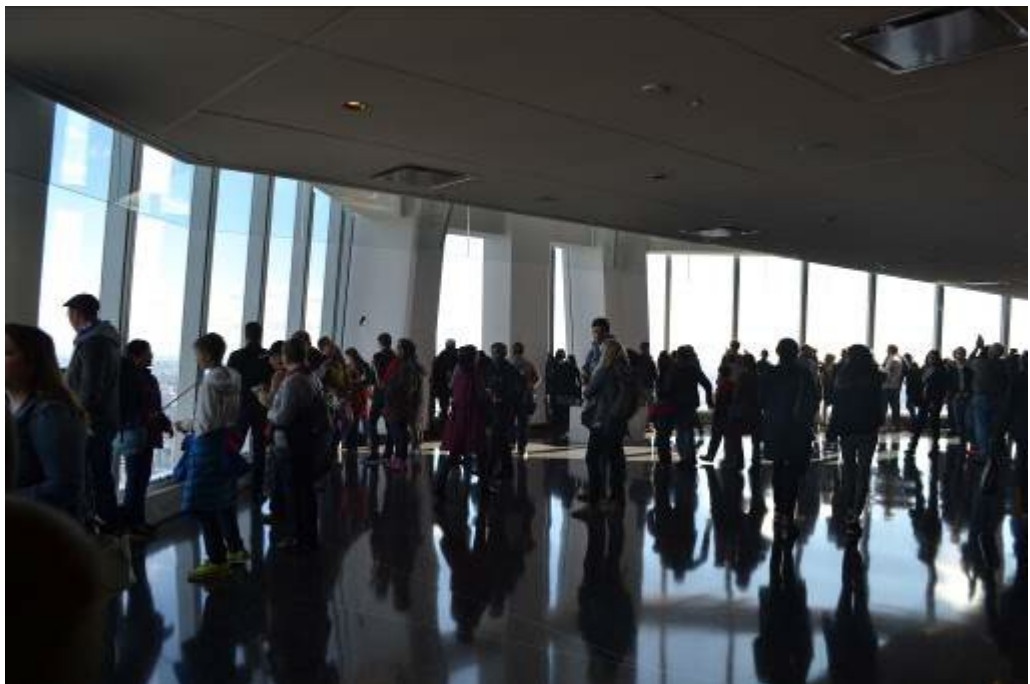

**Figure 4.** Observation Deck on the 102nd Floor of One World Trade Center. Visitors take fast elevators that travel 394 m (1293 ft) from the ground floor to the observation deck in 47 s. (Photo by author).

Eight 2.3-ton electric motors were placed on the roof of One WTC to power the fast elevators. Each elevator is controlled by a pulley-like device that consists of a cab and counterweights connected by a cable. Tenants use an express shuttle to get to the sky lobby on the 64th level, where "local" elevators that travel to higher levels are located. Almost every component of the elevator systems, including the high-speed double-deck elevators, computerized roller guides, air pressure differential systems, destination dispatching systems, and entertaining electronic displays, use cutting-edge technology.

*4.3. Computerized Roller Guides*

It takes more than powerful motors to transport an elevator across long distances swiftly. Like bullet trains, elevators that operate at high speeds require exceptionally smooth tracks and rail junctions. Due to alignment precision requirements, the vertical placement of elevator rails limits their length to about 4.9 m (16 ft), meaning that every elevator path will require a lot of rail joints. The train-rail segments in One WTC have been elongated to decrease the number of joints the cars must pass, making the rides more comfortable for passengers [22].

Since skyscrapers swing slightly day and night due to temperature changes (contraction and expansion), wind forces, and other factors, elevators must account for these minute variations in the spacing between guide rails [23]. ThyssenKrupp designed the One WTC's automatic roller guides to deal with these problems. Innovative roller guides apply

pressure in the opposite direction of motion, dampening the impact of irregularities. The elevator's wheels stay in touch with the rails due to the roller guides moving up and down with the car. The polyurethane rollers at One WTC can endure minor defects in the rail connections. A device that exerts a force against the rails also controls them, reducing the likelihood of tremors and rattling due to alignment issues or other defects. These dynamic roller guide systems, also known as real-time shock absorbers, are intelligent. They act like a driver who has seen a large pothole in the road and swerved slightly to avoid it. If a pothole is on the right side of the road, the driver will make a slight left turn to avoid it, and vice versa. This means that the express elevators are not only fast (25% faster than the express elevators that served the original World Trade Center Twin Towers) but also comfortable for passengers, as there is no shaking or rattling [22] (Figure 5).

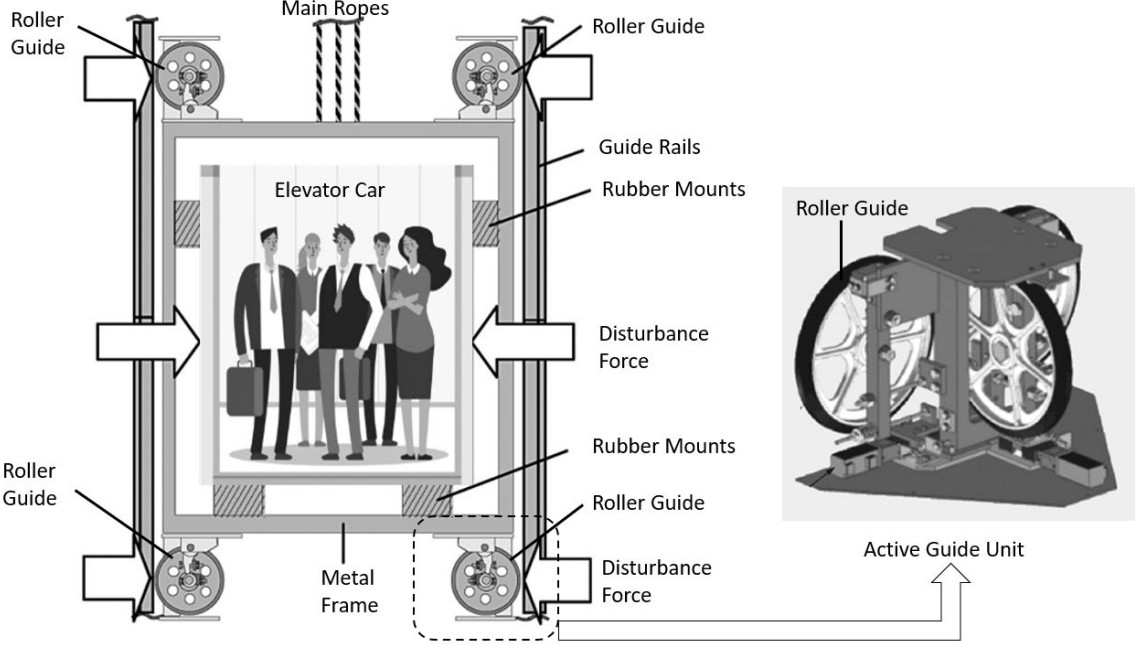

**Figure 5.** Computerized roller guides system. (Source: author).

*4.4. Air Pressure Differential*

The air pressure differential must be considered when designing and constructing high-speed elevator systems that travel long distances, such as those found in very tall buildings. The rapid travel of an elevator across floors causes air resistance problems in the elevator shaft. To illustrate the point, a train approaching a station creates an air-pressure effect by pushing a wall of air in front of it. Similarly, relative air displacement occurs when an average vehicle weighing 4500 kg (9921 lb) and a counterweight weighing 7300 km (16,094 lb) rapidly ascend or descend in an elevator shaft. Due to a high-pressure zone above the cab and a low-pressure zone below it, the hoistway doors above the vehicle are pushed into the hallway, while the hoistway doors below the car are drawn into the hoistway. A solution was devised by ThyssenKrupp, which consisted of attaching wedge-shaped aluminum shrouds to the top and bottom of the cabs to maximize the aerodynamic efficiency of the cabs as they ascend and descend the shafts. Because of its aerodynamic design, the cab has less wind noise, less door rattling, less air displacement, and less resistance to the passage of air [22,23].

The second air-pressure issue is related to the comfort and safety of passengers, notably the "ear-popping" effect that happens when the elevator moves more quickly. Although it happens because of the abrupt and quick shift in air pressure when the elevator ascends and descends, this problem is more noticeable during the descent. It is vital to keep in mind that lifts in very tall structures decline more quickly than commercial aircraft. An airplane's

landing operation can last up to 30 min, giving plenty of time to regulate the cabin's air pressure. As opposed to this, elevators must adjust the air pressure in as little as 30 s. As a result, it takes less time for elevator passengers to adjust. To avoid passengers' ears from popping, ThyssenKrupp invented pressurized vehicles at One World Trade Center (i.e., adding extra air pressure to make up for pressure drops). Engineers at ThyssenKrupp have conducted a thorough analysis to determine the ideal speed for fans that control vehicle air pressure as elevators rapidly descend or ascend. Overall, elevators can only fall at a maximum speed of 10 m per second (33 ft/s) in all cases due to the air pressure problem [22,23].

### 4.5. Structural Safety

One hundred ninety-eight of the largest and fastest elevators in the world could be found at the Twin Towers of the former World Trade Center. However, on 9/11, at least 200 people died in the elevators of these towers, making it the most significant elevator catastrophe in history. The steel exoskeleton framework of the towers, whose weakness was exposed in this attack, was blamed for this issue. In contrast, a 3.3 foot (1 m) thick concrete casing surrounds the elevator hoistways of the One WTC. In case of an emergency, such as a fire, residents are also safeguarded in a refuge room on each floor. Additional steps were taken to increase passenger safety. The emergency elevator typically stops directly below the damaged floor during a fire, leaving the firefighters no choice but to climb the stairs to the fire's source. Firefighters can access the fire levels directly because the emergency elevator shaft at One WTC is kept at negative pressure to prevent smoke from entering. A second door that may be opened in the emergency elevator's cab allows the firefighters to enter a different passageway [24].

### 4.6. Destination Dispatching System and Security

The building employs a Destination Dispatching System (DDS) to offer a faster and more efficient service. The utilized DDSs in 63 of the building's elevators are connected via an Intranet, and passengers headed for the same destination are grouped and share an elevator. Passes to enter the building that reveal the holder's identity and the department they work in are made mandatory to increase safety. The elevator receives that information when visitors swipe their badges at the turnstile. The elevator's number flashes up on the digital screen in real time. There is a touch screen outside the elevators that guests and employees with multiple-floor access can use to override the system and change their destination [25].

### 4.7. Elevator Maintenance

One WTC's elevator maintenance program uses Microsoft's Azure Intelligent Systems Service. By continuously feeding service engineers real-time data, this technology responds to issues before they become serious, enabling them to take action to prevent elevators from breaking down. These data are added to dynamic predictive models, which assist engineers in taking preventative measures. The system immediately offers the most likely causes when an elevator reports a problem. Such a process shortens the potential downtime for the elevators by enabling technicians to identify and start repairs more promptly [24].

### 4.8. Entertainment

The interior walls of the five high-speed elevators that travel to the observatory feature large high-definition monitors designed to resemble windows. During the 47 s ride, the monitors display a historical panorama of the city—a CGI timelapse of New York's development over the past 500 years, starting in 1500. In the 1600s, colonial homes sprout on the island's meadows. As colonial New York takes shape in the 1700s, the shoreline shifts further south. Travelers could see the tidal strait known as the East River by 1839. The old World Trade Center's south tower, built in 1971, briefly appears to rise above Manhattan. As the new tower's steel beams and structural supports eventually spindle

outward into being, the building seems to come together around the elevator just in time for arrival on the 102nd floor [22,26].

## 5. Elevator Modernization and Upgrading Systems

A modernization project may be required for elevator systems that have been in use for ten years or more. The components of an elevator system will be assessed first for their level of safety and compliance with the most recent building code standards. Every single element necessary for the elevator's effective and risk-free operation will be examined throughout the evaluations. Assessments also include an analysis of the necessary engineering support systems and accessibility provisions. Modernization projects should be considered when the lifespan of an existing elevator system must be extended. There are many ways to modernize and upgrade existing elevators. For example, upgrading to LED bulbs can save up to 45% of the energy incandescent bulbs use. Building owners can use digital solutions that collect elevator data to provide information that helps to track a system's energy use. It is also recommended to replace conventional motors with regenerative drives. As was explained earlier, the regenerative drive captures unused energy generated during braking and feeds it back into the building power system. Furthermore, it is recommended to switch to a gearless drive. Due to their smaller size and lack of gears, these machines require much less energy than their geared counterparts. The following explains a few modernization options [27,28].

### 5.1. Full Replacement

This solution is appropriate for older elevators with issues including (1) being out-of-date and requiring a lot of energy to operate; (2) being inaccurately level with landing floors, creating risky tripping hazards; or (3) being unreliable with frequent breakdowns. Whole elevator replacement is a service offered by elevator companies in which the old elevator is entirely removed, and a new, accessible traction elevator is installed inside the building's existing shaft [29].

### 5.2. Traction Modernization

If the elevator: (1) uses a lot of electricity; (2) does not level precisely with landing floors; or (3) has an old-fashioned interior, modular upgrading is the best option. Replacing systems, such as hoisting equipment, signalization, or doors, can improve the performance and aesthetics of the elevator and possibly reduce running costs [29].

### 5.3. Hydraulic Modernization

This option is ideal if the building's hydraulic elevator: (1) is frequently out of service due to mechanical wear and tear and worn-out or obsolete electrical components; (2) operates inconsistently and unreliably, resulting in increased maintenance costs; or (3) has reached the end of its useful life. The hydraulic elevator renovation solution provides the building with a modern elevator that offers a quiet, smooth ride and complies with the most recent accessibility and safety requirements [29].

### 5.4. Smartphones and Wearable Devices

Some elevator companies have developed smartphone applications that enable users to call elevators. These applications eliminate the need to press any elevator buttons, allowing anyone, with or without a disability, to access elevators quickly, safely, and comfortably. ThyssenKrupp has developed the AGILE app that enables tenants to operate elevators via their smartphone or wearable device. The app, compatible with Android and iOS devices, mitigates elevator traffic congestion and physical interaction with elevator surfaces, buttons, and handrails. Similarly, Nayar Systems has developed the Pulse elevator smartphone app [30].

*5.5. Holographic Elevator Buttons*

Another innovation on the horizon is a system of holographic buttons, which avoids the need to touch the elevator buttons. Inspired by sci-fi, the new technology offers a floating panel with options for users to select by pressing or pointing to buttons on a holographic image without touching any buttons. Alternatively, passengers can use simple voice commands. These revolutionary designs provide convenience as well as security and safety in the face of the global COVID-19 pandemic [31].

*5.6. Ultraviolet Rays That Disinfect Elevators*

On the market today is a new type of intelligent elevator that uses UV radiation to keep the cabin clean. As a precaution against COVID-19, the mechanism will eliminate any remaining traces of viruses and bacteria as soon as the elevator stops, and no one is present [32].

*5.7. Integrated Elevator Access Control Security*

Offering users touchless operating panels and access through destination dispatch systems are excellent approaches to boost convenience and security in office buildings. For example, using IoT-connected equipment with Openpath's elevator access control and braXos's destination control system, building users can experience touchless elevator access and improved security. An integrated elevator security system can lock down the elevators immediately in case of a fire alarm and instantly alert security officers and first responders. This security system is made by Motorola Solutions, headquartered in Chicago, IL, USA. Further, a visitor experience is made more straightforward and practical. A registered visitor, contractor, or vendor would immediately receive a digital guest pass for the floors and period they requested to stay after checking in. Overall, supervising guest visitors on the premises using these systems increases security [33–35].

**6. Key Future Developments**

*6.1. Circulating Multi-Car Elevator System*

With conventional elevators, a single vehicle ascends and descends the same elevator shaft. In a circulating multi-car elevator system, however, numerous cars (each equipped with a revolving magnetic array propulsion wheel) travel in a circular motion within the span of two traditional elevator shafts (Figure 6). This system is like a Ferris wheel, except each car has its motor instead of counterweights. Steel ropes drive its prototype for revolving multi-car elevators. This method attaches two cars to two circulating steel ropes to form a unit. The support rail system fitted throughout the elevator shaft guides the movement of the cars and prevents lateral swaying. Hitachi verified this elevator system using a replica one-tenth of the actual size. The spinning multi-car elevator is anticipated to increase capacity, reduce the number of shafts necessary, and shorten waiting times. Sadly, the existing prototypes require additional safety enhancements to fulfill international standards [36–38].

*6.2. Multi-Directional Elevators*

Conventional elevators move a cabin strictly up and down using cables. In contrast, the MULTI manages elevators more like a subway system. The cabins shift horizontally, vertically, frontally, and rearwards. Each system comprises multiple cabins, each powered by a linear induction motor. Inspired by the TWIN system, in 2017, ThyssenKrupp produced the first multi-car ropeless elevator system. Instead of ropes, a motor system pushes each car around a looping shaft. This system is no longer height-restricted due to the lack of ropes necessary to move the cars (Figure 7). As a result, the MULTI provides a method to boost elevator height above the KONE's 650 m (2133 ft) steel rope limitation. Because the MULTI utilizes many vehicles, it guarantees a shorter wait time. Finally, it promises to cut energy consumption by incorporating a "smart" device that reduces peak power demand by up to 60 percent, decreasing the entire structure's carbon footprint [39–41].

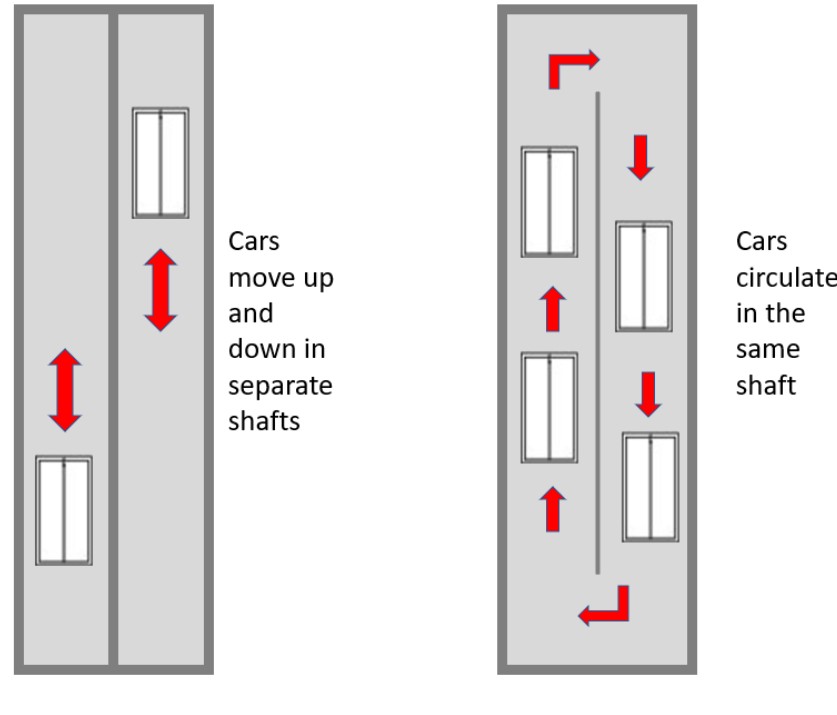

**Figure 6.** Conventional Elevator System versus Circulating Multi-Car Elevator System. (Source: author).

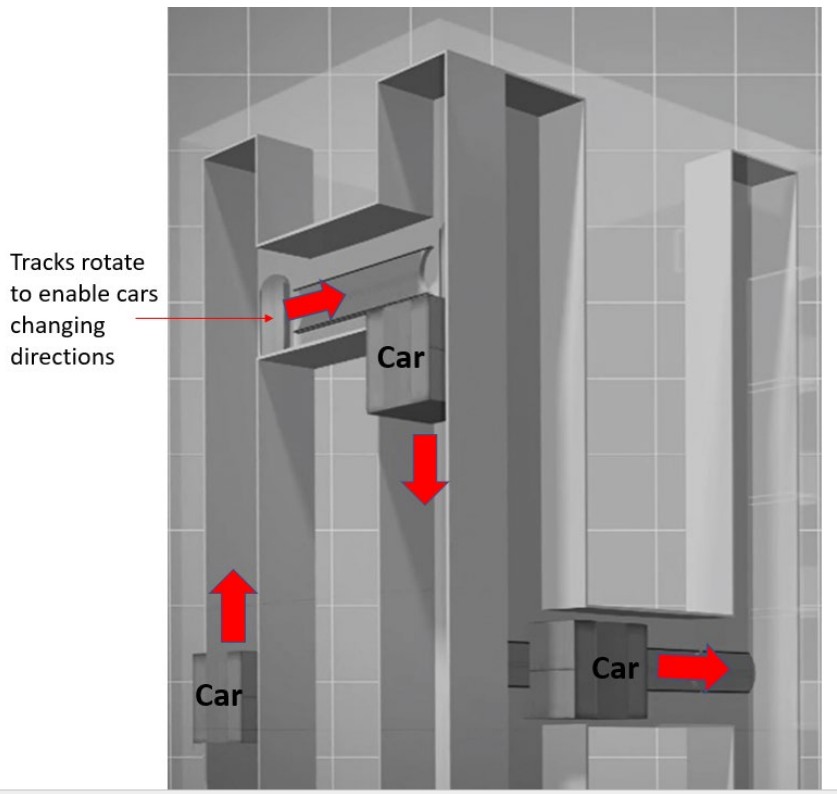

**Figure 7.** Multi-Directional Elevators. (Source: author).

### 6.3. Virtual Reality Diagnostics

The use of virtual reality headsets could revolutionize the way that elevator technicians execute their jobs. VR headsets allow them to diagnose elevators' issues without risking their lives by having to climb multiple stories or hang from a lift shaft. By putting on the glasses and utilizing hand gestures, technicians can examine the entire engine and its components in greater detail. The maintenance crew can also expand the view to carrying out a "virtual walk-through" to determine what parts need to be fixed or replaced.

## 7. Summary

This article reviews milestone elevator system developments. Overall, with continuous technological advancements, elevators will be able to meet the requirements of buildings with extreme heights. Digital innovations will support urbanization well into the foreseeable future, making it easier for passengers to travel across buildings conveniently and safely. Sustained collaboration between engineers, architects, computer scientists, elevator manufacturers, and builders may lead to cost-effective solutions that improve performance and promote efficiency [42,43]. The following offers essential highlights.

### 7.1. Escalating towards a Greener Future

The vertical mobility endeavor has the motivation to embrace environmental consciousness, reduce the negative impact on the environment, and save energy. For example, the regenerative drive is a technology that helps minimize energy usage by collecting energy that would have been lost otherwise and sending it back to the building power system. Further, buildings are implementing LED lighting solutions and using standby modes in elevators to reduce energy consumption. The use of programmable dispatch software helps reduce the necessary number of stops, which in turn saves energy. In addition, energy consumption is significantly reduced compared to older elevator control systems due to the use of microprocessor-based controls in modern elevator systems. Finally, the market currently provides elevators that have machines that are not only more compact but also more robust. This helps conserve space and energy [44–46]. The main features of green elevators include:

- Machine-Room-Less (MRL) system;
- Gearless traction motor;
- Drive systems that regenerate energy;
- Computerized precision traffic control that optimizes the performance of a group of elevators and decreases light-load trips;
- In-cab sensors and software that make the elevator go into sleep mode when not in use, turning off the music, video, lighting, and ventilation;
- Destination dispatch control software to improve passenger traffic flow (see next section).

### 7.2. Efficient Movement of People and Algorithms

The destination dispatch technology is being adopted to move people efficiently throughout the building. This is accomplished by grouping passengers traveling to the same destination floor. Just entering their destination into a control panel is what passengers need to do. In addition, after the epidemic, this technology was hailed as the solution to the congestion problem in elevator cars and elevator lobbies to meet the demand for social separation [47,48]. Almost every leading elevator company is working on cutting-edge controller systems. Elevators will likely not have any up and down buttons, but riders will have to enter their floor numbers before walking into elevators using their hand-held and wearable devices. The elevator system can reduce wait and transit times by using the collected data to make smarter decisions about which elevators should pick up which passengers and on what floors.

### 7.3. Technologies That Aid in Hygiene

The recent pandemic has altered our travel and building navigation. One of the most remarkable inventive technologies consists of buttons that do not need to be touched to operate, reducing the spread of infections and disease. Users with contactless elevator buttons can summon a lift simply by holding their finger one to three centimeters away from the panel on which the buttons are located. This feat is possible due to the employment of an infrared sensor interface. Touchless solutions such as Otis eCall$^{TM}$ (produced by Otis, a global elevator company headquartered in Farmington, CT, USA) and Schindler's ElevateMe (produced by Schindler, a global elevator company headquartered in Ebikon, Switzerland) help reduce public elevator button use. Similarly, passengers can send an elevator call from their smartphones using an app, reducing the need to touch elevator buttons. Elevator air purifiers and escalator handrail sanitizers can improve building hygiene [47,49].

### 7.4. Remote On-Time Services

Elevators require regular maintenance. Thus, the elevator industry uses remote monitoring systems to aid building authorities in diagnosing and evaluating an elevator's performance. This is to ensure the continued functionality of elevators. Internet of Things technology assists in delivering real-time data, providing insights to avoid unscheduled shutdowns, and facilitating problem prevention. To improve customer service, leading elevator manufacturers such as Otis are equipping their field staff with cell phones to monitor, predict, and remedy any elevator problems remotely [43,50].

### 7.5. The MULTI System

Some manufacturers are pushing the edge by designing elevators that can travel vertically, diagonally, and horizontally across a building, although most elevators can only move in one direction, often vertically. Even though this concept is currently in the testing phase, it can potentially revolutionize how elevators are integrated into buildings. Because there are no ropes, the motors move the car's weight, reducing used energy. Elevators moving in vertical and horizontal directions without height constraints and passing one another in adjacent shafts will allow for much greater architectural freedom [49,51].

### 7.6. Speed

Elevator speed has improved remarkably in the recent past. However, research indicates that elevator technology may be getting dangerously close to the speed limits that human riders can tolerate. Despite efforts to make for a comfortable ride, such as air conditioning, pressurized cabs, and decreased travel vibration, faster speeds may make humans sick. So, other development areas will soon become more important than pure speed to engineers and elevator firms. These efforts include purified air circulation systems, pressurized cabs, efficient evacuation, decreased vibration, and, most importantly, reduced manufacturing costs and maintenance [52].

**Funding:** This research received no external funding.

**Institutional Review Board Statement:** Not applicable.

**Data Availability Statement:** Not applicable.

**Acknowledgments:** The author deeply thanks the editors and reviewers for insightful comments.

**Conflicts of Interest:** The authors declare no conflict of interest.

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
