# Peer review of "Elevator Technology Improvements: A Snapshot"

_encyclopedia, doi:10.3390/encyclopedia3020038_

Round 1

Reviewer 1 Report

Comments and Suggestions for Authors

The paper delivers what it promises in the abstract. However, the style of writing appears bombastic at places.  

There are a few sentences which require rewriting.  Best would be for the author to ask a native English speaker to carefully read the manuscript, and suggest improvements.  

EXAMPLES:  

25 Some builders, building owners, and tenants may have personalities that push them to make taller than their competitors or have their condo or business address in a supertall structure.

"Like smartphones and televisions, elevators are becoming increasingly high-tech"  Compare with automobile or rail transport instead of smartphones

"Unlike conventional hydraulic lifts, modern electric elevators significantly cut back on power consumption and have prevailed."  Sentence appears incomplete     

Table 1. Global Top Ten Elevator Companies. (Source: elevatorworld.com) KEEP WITH NEXT

405 "Moving an elevator quickly and over great distances requires more than just strong motors and current." replace "strong motors and current with powerful motors 

620 7.1. Escalating Towards a Greener Future KEEP WITH NEXT

657 "drastically reducing the spread of infections and disease" drastically appears an exaggeration, as many other factors contribute to the spread of  SARS-COVID 

REFERENCES: 

"Fleischmann, C., Scherag, A., Adhikari, N. K., Hartog, C. S., Tsaganos, T., Schlattmann, P., ... & Reinhart, K."    What is .... ?

Author Response

Reviewer #1

Thank you very much for your helpful feedback. I agree with your comments. My answers to specific remarks are in red.  

The paper delivers what it promises in the abstract. However, the style of writing appears bombastic at places.  

I agree. I have rewritten a good portion of the paper.

There are a few sentences which require rewriting.  Best would be for the author to ask a native English speaker to carefully read the manuscript, and suggest improvements.  

I also sought help from a native English speaker.

EXAMPLES:  

25 Some builders, building owners, and tenants may have personalities that push them to make taller than their competitors or have their condo or business address in a supertall structure.

I deleted and replaced the whole sentence.

"Like smartphones and televisions, elevators are becoming increasingly high-tech"  Compare with automobile or rail transport instead of smartphones

I replaced them with the suggested words.

"Unlike conventional hydraulic lifts, modern electric elevators significantly cut back on power consumption and have prevailed."  Sentence appears incomplete   

I rewrote this sentence and the following sentence.

Table 1. Global Top Ten Elevator Companies. (Source: elevatorworld.com) KEEP WITH NEXT

Corrected

405 "Moving an elevator quickly and over great distances requires more than just strong motors and current." replace "strong motors and current with powerful motors 

I replaced it accordingly.

620 7.1. Escalating Towards a Greener Future KEEP WITH NEXT

Corrected

657 "drastically reducing the spread of infections and disease" drastically appears an exaggeration, as many other factors contribute to the spread of  SARS-COVID

I deleted “drastically.”

REFERENCES: 

"Fleischmann, C., Scherag, A., Adhikari, N. K., Hartog, C. S., Tsaganos, T., Schlattmann, P., ... & Reinhart, K."    What is .... ?

I replaced this reference with a more relevant one.

Also, I replaced several references with others that are more closely relevant. These are highlighted in yellow.

----------------------------------------------------------------------------------------------

Reviewer 2 Report

Comments and Suggestions for Authors

This is an interesting article about effective vertical mobility in tall buildings and how advances in elevator technology affect their construction and operation. The article examines developments such as smart elevators, ropeless MULTI elevators, AI, IoT and extended reality technologies. It also analyzes the use of these technologies in large-scale projects such as the One World Trade Center in New York and their response to the COVID-19 pandemic.

The author has taken some parts of the text from the other article of his authorship without citing it. I consider that these parts should be rewritten and the author should also cite the original source. This article is:

Al-Kodmany, K., 2023. Smart Elevator Systems. Journal of Mechanical Materials and Mechanics Research. 6(1): 41-54. DOI: https://doi. org/10.30564/jmmmr.v6i1.5503

The author should review the following aspects:

1. All the units must be in the SI.

2. Some Figures and Tables are numbered incorrectly. Check that the reference in the text and the Table header matches. Check the order of numbering.

3. Reconsider heading 2.4. I think it is interesting to talk about the different companies but not making a ranking.

4. lines 469-471: "...negative pressure..." Is this correct?

5. line 361: "One World Trade Center, New York, USA - Is it a heading? If it is a heading, you must number it.

Author Response

Reviewer #2

Thank you very much for your helpful feedback. I agree with your comments. My answers to specific remarks are in red.  

This is an interesting article about effective vertical mobility in tall buildings and how advances in elevator technology affect their construction and operation. The article examines developments such as smart elevators, ropeless MULTI elevators, AI, IoT and extended reality technologies. It also analyzes the use of these technologies in large-scale projects such as the One World Trade Center in New York and their response to the COVID-19 pandemic.

The author has taken some parts of the text from the other article of his authorship without citing it. I consider that these parts should be rewritten and the author should also cite the original source. This article is:

Al-Kodmany, K., 2023. Smart Elevator Systems. Journal of Mechanical Materials and Mechanics Research. 6(1): 41-54. DOI: https://doi. org/10.30564/jmmmr.v6i1.5503

I have included this reference in the reference list, reference 2. Also, I mentioned it up front in the text in the second sentence of the Abstract. Simultaneously, I rewrote much of the shared content in the above reference.

The author should review the following aspects:

  1. All the units must be in the SI.

I rewrote all units according to the SI. I also prioritized SI, mentioning it first, and then included the equivalence in the British system, facilitating convenience for the readers.

  1. Some Figures and Tables are numbered incorrectly. Check that the reference in the text and the Table header matches. Check the order of numbering.

They are now corrected.

  1. Reconsider heading 2.4. I think it is interesting to talk about the different companies but not making a ranking.

I rewrote the heading “Leading……” and rewrote the content of this section so that it does not rationalize ranking but highlights the significance of these global companies.

Also, I rewrote the table title as “Leading….” and removed from it the “Rank.” 

  1. lines 469-471: "...negative pressure..." Is this correct?

Yes.

  1. line 361: "One World Trade Center, New York, USA - Is it a heading? If it is a heading, you must number it.

Corrected -- I placed it as a continuation of the Case Study heading.

Round 2

Reviewer 1 Report

Comments and Suggestions for Authors

The author has thoroughly reviewed the manuscript which has now achieved journal publication standard. Figures 1 and 4 appear to have low resolution, and could be replaced with better ones.  

Author Response

Reviewer # 1

The author has thoroughly reviewed the manuscript which has now achieved journal publication standard. Figures 1 and 4 appear to have low resolution, and could be replaced with better ones.  

--------------------------------------------------

Thank you for the compliment.

I have improved the resolution of Figure 1. Also, I will hand over the high-resolution image when we get to the production phase.

Also, I have replaced Figure 4 with a newer, good-resolution one.

Reviewer 2 Report

Comments and Suggestions for Authors

Dear author,

I recommend rejecting the manuscript because I still find too many similarities with the aforementioned article (Smart Elevator Systems) published by you. Furthermore, this manuscript contains four Figures and one Table taken from that paper, which are not correctly cited.

Author Response

Reviewer # 2

I recommend rejecting the manuscript because I still find too many similarities with the aforementioned article (Smart Elevator Systems) published by you. Furthermore, this manuscript contains four Figures and one Table taken from that paper, which are not correctly cited.

---------------------------------------------

I have rewritten much of the overlap and repetition; please see the new version. The report shows a 13% rate, and I hope now it is reduced to an acceptable rate. All the rest of the overlaps/repetitions are 1% each (except one 4%), making them insignificant. Admittedly, substantial technical terms and engineering facts are tough to paraphrase, fearing changing their conventional meanings. 

I have replaced these figures with new ones.

I suggested removing the second table since it is not critical to the paper.

Round 3

Reviewer 1 Report

Comments and Suggestions for Authors

The paper is acceptable in its current form. 

Reviewer 2 Report

Comments and Suggestions for Authors

Dear autor,

I appreciate the effort you have made in revising the manuscript and I find it appropriate that you have rewritten the parts that had similarities with your previously published paper. I also consider it appropriate that you have modified the figures and deleted Table 2.

On the other hand, I think it is not necessary to indicate in the figure captions that the figure is of your own authorship and, therefore I would delete: source: author.

Kind regards,